# miR-30c-5p Gain and Loss of Function Modulate Sciatic Nerve Injury-Induced Nucleolar Stress Response in Dorsal Root Ganglia Neurons

**DOI:** 10.3390/ijms252111427

**Published:** 2024-10-24

**Authors:** Raquel Francés, Jorge Mata-Garrido, Miguel Lafarga, María A. Hurlé, Mónica Tramullas

**Affiliations:** 1Departamento de Fisiología y Farmacología, Facultad de Medicina, Universidad de Cantabria, 39011 Santander, Spain; raquelfrances1@gmail.com (R.F.); hurlem@unican.es (M.A.H.); 2Instituto Marqués de Valdecilla (IDIVAL), 39011 Santander, Spain; jorge.mata-garrido@pasteur.fr (J.M.-G.); miguel.lafarga@unican.es (M.L.); 3Departamento de Anatomía y Biología Celular, Universidad de Cantabria, 39011 Santander, Spain; 4Centro de Investigación Biomédica en Red Sobre Enfermedades Neurodegenerativas (CIBERNED), 28029 Madrid, Spain

**Keywords:** neuropathic pain, nerve injury, miRNAs, miR-30c, chromatolysis, nucleolar stress, dorsal root ganglia, nucleolus, Cajal body

## Abstract

Neuropathic pain is a prevalent and debilitating chronic syndrome that is often resistant to treatment. It frequently arises as a consequence of damage to first-order nociceptive neurons in the lumbar dorsal root ganglia (DRG), with chromatolysis being the primary neuropathological response following sciatic nerve injury (SNI). Nevertheless, the function of miRNAs in modulating this chromatolytic response in the context of neuropathic pain remains unexplored. Our previous research demonstrated that the intracisternal administration of a miR-30c mimic accelerates the development of neuropathic pain, whereas the inhibition of miR-30c prevents pain onset and reverses established allodynia. In the present study, we sought to elucidate the role of miR-30c-5p in the pathogenesis of neuropathic pain, with a particular focus on its impact on DRG neurons following SNI. The organisation and ultrastructural changes in DRG neurons, particularly in the protein synthesis machinery, nucleolus, and Cajal bodies (CBs), were analysed. The results demonstrated that the administration of a miR-30c-5p mimic exacerbates chromatolytic damage and nucleolar stress and induces CB depletion in DRG neurons following SNI, whereas the administration of a miR-30c-5p inhibitor alleviates these effects. We proposed that three essential cellular responses—nucleolar stress, CB depletion, and chromatolysis—are the pathological mechanisms in stressed DRG neurons underlying neuropathic pain. Moreover, miR-30c-5p inhibition has a neuroprotective effect by reducing the stress response in DRG neurons, which supports its potential as a therapeutic target for neuropathic pain management. This study emphasises the importance of miR-30c-5p in neuropathic pain pathogenesis and supports further exploration of miRNA-based treatments.

## 1. Introduction

Pain is an unpleasant sensory experience that alerts the organism to real or potential damage, and it usually disappears after healing [1]. However, pain can become chronic and persist for months or even years, even though the original injury may have long since disappeared. Chronic pain affects millions of people around the world and constitutes a global health issue with a significant socio-economic impact [2]. Neuropathic pain is a type of chronic pain caused by a lesion or disease of the somatosensory nervous system and affects almost 10% of the global population [3,4]. The conditions and pathophysiological states that determine the onset of neuropathic pain include metabolic disorders (e.g., peripheral diabetic neuropathy), traumatic damage to the nervous system (e.g., spinal cord injury and amputation), viral infections (e.g., post-herpetic neuralgia), chemotherapy-induced peripheral neuropathies, and autoimmune disorders affecting the central nervous system (e.g., multiple sclerosis) [5]. Neuropathic pain is typically associated with symptoms such as spontaneous pain, pain evoked by innocuous stimuli (allodynia), exaggerated pain to noxious stimuli (hyperalgesia), and even paresthesia (abnormal sensation of burning, numbness, tingling, itching, and pricking or prickling) [6,7]. The treatment of neuropathic pain remains a significant challenge due to the current limitations of available options, which primarily focus on symptom suppression and are often associated with low efficacy and undesirable side effects [8,9]. Consequently, a deeper understanding of the underlying mechanisms of neuropathic pain is crucial for the development of novel and effective therapeutic strategies for long-lasting pain management.

Neural damage causes long-lasting maladaptive plastic changes along the sensory pathways, from the peripheral to the central nervous system. Structural plasticity leads to hyperexcitability and spontaneous activation of the nociceptive pathway, which may contribute to the generation, development, and maintenance of neuropathic pain [10]. Traumatic injuries (crush, section, ligature) to the sciatic nerve are widely used to induce neuropathic pain in rodents. Damaged first-order nociceptive neurons in the lumbar dorsal root ganglia (DRG) are responsible for the retrograde signals from injured axons to the second-order neuron in the dorsal horn of the spinal cord (SDH), which ultimately primes the nociceptive pathway to neuropathic pain development. Following axon damage, several cellular and molecular stress responses are initiated in the neuronal soma, which is thought to be responsible for either nerve repair or neuronal dysfunction depending on the particular insults, contexts, etc. [7].

Chromatolysis represents a significant neuropathological phenomenon that emerges in response to a range of insults, including traumatic injury, ischemia, metabolic diseases, stress, and toxicity that occurs in the cell body of damaged neurons [11,12,13,14,15,16]. It is characterised by the disruption, dispersal, and redistribution of the cytoplasmic protein synthesis machinery and Nissl bodies (NBs), which are composed of arrays of rough endoplasmic reticulum (RER) cisterns and free polyribosomes [17]. A clearing of cytoplasm and frequent accumulation of neurofilaments reflect this neuronal reactive response. Specifically, “central chromatolysis”, the most common form of chromatolysis, is characterised by a loss or complete absence of NBs from the centre of the cell body and peripheral displacement of the nucleus [11,12,13,14,15,16]. Following injury, neurons can recover after the activation of neuroprotective mechanisms that promote neuronal survival. However, chromatolysis is more often a precursor of apoptosis [18].

The nucleolus is a nuclear compartment that plays an essential role in ribosome biogenesis. This complex process encompasses the synthesis, processing, and maturation of ribosomal RNAs (rRNAs) and their assembly with ribosomal proteins into ribosomal subunit particles [19,20,21]. Furthermore, the nucleolus has been identified as a central sensor and mediator of various cellular stresses, introducing the concept of “nucleolar stress” (NS) [22,23]. Consequently, the NS response detects perturbations in nucleolar functions, such as inactivation of RNA polymerase I transcription, impairment of both rRNA processing and ribosome assembly, and abnormal accumulation of ribosome-free ribosomal proteins, which lead to disturbances in protein synthesis and cell homeostasis [22,23,24].

Compelling evidence obtained from animal models supports that chromatolysis and NS are pathological hallmarks of several neurodegenerative and neurodevelopmental disorders [25,26,27,28,29,30,31,32,33,34], essentially due to the higher protein synthesis demand of neurons in comparison with other cell types [35,36]. In the case of DRG neurons, several studies have reported the incidence of a chromatolytic response following peripheral nerve injury, such as transection of the sciatic nerve, which is dependent on the severity of the damage [37,38,39]. Nevertheless, to the best of our knowledge, there are no studies that directly address the relationship between the stress response of DRG neurons following injury and the subsequent development of neuropathic pain. The mechanisms underlying neuropathic pain are complex and involve molecular and cellular changes in the nervous system. In this regard, our group has focused on the role of miRNA-related mechanisms involved in the neural plasticity triggered by peripheral nerve injury, which contributes to neuropathic pain development. Our previous results support a pivotal etiopathogenic role for miR-30c-5p in neuropathic pain development, maintenance, and resolution. In rodent models, SNI induces miR-30c overexpression in the SDH, DRG, and circulating fluids, including the cerebrospinal fluid (CSF) and plasma, which correlates directly with the severity of the allodynia developed. Interestingly, the intracisternal administration of a miR-30c-mimic accelerates the development of neuropathic pain, whereas silencing miR-30c with an inhibitor prevents pain development and reverses fully established allodynia [40]. Furthermore, the long-lasting antiallodynic effect induced by a miR-30c-5p inhibitor is associated with strong global DNA hypermethylation in DRG and SDH neurons, accompanied by the upregulation of DNA methyltransferases [41].

We postulated that the DRG response to stress induced by nerve injury may be involved in neuropathic pain development through a mechanism mediated by miR-30c-5p and that targeting miR-30c-5p may have therapeutic implications. The present study thus aimed to assess the cellular basis of miR-30c-5p gain and loss of function in DRG neurons following SNI, particularly the potential relationship between NS and chromatolysis and neuropathic pain.

## 2. Results

### 2.1. Administration of a miR-30c-5p Mimic Aggravates the Chromatolytic Damage in DRG Neurons After SNI, While Injection of a miR-30c-5p Inhibitor Reduces It

NBs represent a major neuronal component of the protein synthesis machinery [17]. Axotomy, as well as other neuronal injuries, such as ischemia, stress, and toxins, can trigger chromatolysis that normally leads to apoptosis unless a reparation process is initiated [18,42].

Therefore, we first explored the alterations in the integrity and distribution of NBs after SNI and the effects of modulating the levels of miR-30c-5p in vivo. For that purpose, mechanically dissociated neurons from the L4, L5, and L6 DRGs were processed. We performed double fluorescence labelling with propidium iodide (PI)—a cytochemical marker of nucleic acids that preferentially stains rRNA-rich structures in neurons [43]—and immunolabelling for Lamin B1 as a marker of the nuclear envelope. As expected, PI staining in DRG neurons from sham-control rats treated with vehicle (Figure 1A,E), miR-30c-5p mimic (Figure 1B), or miR-30c-5p inhibitor (Figure 1F) revealed round nuclei, prominent rRNA-rich nucleoli, and numerous NBs typically distributed throughout the cytoplasm. In contrast, injured DRG neurons from rats sacrificed on day 5 (Figure 1C) or day 10 (Figure 1G) exhibited strong central chromatolysis with a peripheral displacement of the nucleus. NBs underwent dissolution, and the remaining protein synthesis machinery shifted from the centre to the periphery of the cell body. No significant differences were observed in the integrity and distribution of NBs in DRG neurons when comparing day-5 SNI rats with day-10 SNI rats (*p* = 0.36). DRG neurons from SNI rats treated with miR-30c-5p mimic also exhibited severe chromatolysis and nuclear eccentricity with deformed oval-shaped nuclei when compared with untreated SNI rats (Figure 1D). Remarkably, SNI rats treated with miR-30c-5p inhibitor showed an NB density and distribution pattern comparable to that of sham control rats (Figure 1H).

The percentage of chromatolytic neurons was quantified in 1000 neurons per rat (n = 3 rats per group). Approximately 24% of the DRG neurons from SNI rats showed chromatolysis (Figure 1I). The treatment with miR-30c-5p mimic significantly increased the percentage of chromatolytic neurons (*p* < 0.001) (Figure 1I). Interestingly, the proportion of chromatolytic neurons was significantly reduced when treated with miR-30c-5p inhibitor (*p* < 0.01) (Figure 1J). Furthermore, the proportion of neurons with eccentric nuclei increased significantly in SNI rats treated with miR-30c-5p mimic when compared with SNI vehicle rats (*p* < 0.05, Figure 1K). However, treatment with miR-30c-5p inhibitor prevented the reactive chromatolytic response observed after SNI in rats. Thus, less than 1% of neurons from SNI rats treated with miR-30c-5p inhibitor showed nuclear eccentricity (*p* < 0.01, Figure 1H,L).

Next, we used transmission electron microscopy to further examine the ultrastructural features of SNI-induced chromatolysis in DRG neurons and its modulation by miR-30c-5p interference. Electron microscopy of DRG neurons from sham and SNI rats treated with miR-30c-5p inhibitor confirmed the typical organisation of the cytoplasm with prominent NBs, composed of aggregates of free polyribosomes and stacks of RER cisterns and intercalated bundles of neurofilaments (Figure 2A,C). In contrast, DRG neurons of SNI rats and SNI rats treated with miR-30c-5p mimic exhibited extensive cleared cytoplasmic areas lacking NBs and free polyribosomes, indicating severe chromatolysis (Figure 2B,D). Moreover, chromatolytic regions appeared enriched in neurofilaments and mitochondria. Mitochondria were of variable size; however, very small (<0.5 µm) round mitochondria were frequently observed, suggesting dysfunction in mitochondrial fission dynamics, which can affect the bioenergetic properties of DRG neurons under neuropathic conditions (Figure 2D, insert).

In conclusion, the reactive chromatolytic response of DRG neurons reflects a severe disruption of the protein synthesis machinery and proteostasis after SNI, which is aggravated by treatment with miR-30c-5p mimic. Moreover, inhibition of miR-30c-5p seems to be an efficient neuroprotective mechanism to reduce neuronal damage following SNI.

### 2.2. Administration of a miR-30c-5p Mimic Aggravates the Nucleolar Stress Response in DRG Neurons After SNI, While Injection of a miR-30c-5p Inhibitor Prevents It

The nucleolus plays a pivotal role in the synthesis and processing of rRNAs and the assembly of pre-ribosomal subunits. The function of the nucleolus is reflected in the nucleolar size and structural organisation and distribution of its tripartite components: the fibrillar centre (FC), the dense fibrillar component (DFC), and the granular component (GC) [20,21,22]. Given the essential role of ribosome biogenesis in sustaining protein synthesis activity, we analysed the response of the nucleolus to severe chromatolysis in the DRG of SNI rats and the impact of miR-30c-5p gain and loss of function on the organisation and function of this organelle.

To determine whether the dysfunction in protein synthesis caused by severe chromatolysis modified the nucleolar architecture of DRG neurons, we performed immunolabelling for UBF (upstream binding factor), a marker of FCs. Previous studies have shown that FC number and organisation in mammalian neurons correlate with the transcriptional activity of ribosomal genes [35,44]. Thus, FCs concentrate transcription machinery components, like UBF and RNA polymerase I, which are directly involved in the regulation of rRNA gene transcription [20,21,45]. Double immunolabelling for UBF and Lamin B1 revealed that control sham rats administered with the vehicle, miR-30c-5p mimic, or miR-30c-5p inhibitor exhibited numerous small-sized UBF-positive spots distributed throughout the nucleolar body (Figure 3A,B,E,F). However, a profound reorganisation of UBF-positive FCs was observed 5 and 10 days after SNI (Figure 3C,G), without significant differences between these two groups (=0.33), and aggravated by treatment with miR-30c-5p mimic (Figure 3D). This consisted of the segregation of the numerous small-sized UBF-positive foci of the unaffected nucleolus into one, or rarely a few, large masses as giant FCs. This nucleolar alteration suggests a severe perturbation of RNA polymerase I transcription, a phenomenon characteristic of the NS response [13,23]. Interestingly and consistent with our previous results, SNI rats treated with miR-30c-5p inhibitor preserved the normal organisation of FCs found in control rats (Figure 3H).

The quantitative analysis of the proportion of DRG neurons with segregated UBF-positive giant FCs revealed a significant increase (*p* < 0.01) in DRG neurons from day-5 SNI rats treated with miR-30c-5p mimic when compared with both SNI and Sham rats treated with vehicle (Figure 3I). Importantly, the proportion of DRG neurons with segregated FCs significantly decreased (*p* < 0.001) in day-10 SNI rats treated with miR-30c-5p inhibitor in comparison with SNI rats treated with vehicle (Figure 3J). It is noteworthy that treatment with miR-30c-5p or miR-30c-5p inhibitor in sham rats did not significantly change the typical organisation of FCs in numerous small-sized foci, characteristic of DRG neurons [43] (Figure 3B,F,I,J).

To further understand the potential neuroprotector effect of the miR-30c-5p inhibitor on the nucleolar organisation of DRG neurons from SNI rats, we performed an ultrastructural analysis. Both DRG neurons from sham and SNI rats treated with miR-30c-5p inhibitor exhibited the typical nucleolar configuration of this neuronal type, which is characterised by the presence of numerous small FCs surrounded by a shell of DFC and variable GC masses (Figure 4A,C). Remarkably, severe alterations of the nucleolar structure were commonly found in DRG neurons from SNI rats treated with vehicle or miR-30c-5p mimic (Figure 4B,D). They consisted of the segregation of fibrillar components (FCs and DFC) and GCs in large masses at the nucleolar periphery, indicating a severe nucleolar dysfunction with NS.

In conclusion, the organisation of nucleolar components in SNI rats treated with miR-30c-5p inhibitor is consistent with a neuroprotective effect on nucleolar functions, preventing the NS response.

### 2.3. Effects of miR-30c-5p Modulation on the Cajal Bodies of DRG Neurons After SNI

Cajal bodies (CBs) are nuclear compartments involved in the biogenesis of both small nuclear ribonucleoproteins (snRNPs) and small nucleolar ribonucleoproteins (snoRNPs) required for pre-mRNA and pre-rRNA processing, respectively [46,47]. CBs are transcription-dependent organelles whose number positively correlates with the transcriptional rate of neurons [35,36,45]. Indeed, neurons can accommodate changes in the demand for pre-mRNA processing by regulating the number and size of CBs as well as the transcriptional rate of ribosomal genes [30,35,36,43,48]. The nucleolus and CBs have a close functional relationship, which is reflected in their frequent physical association [45,49]. Specifically, the CB associates with the nucleolar DFC, facilitating the transfer of snoRNPs from CBs to the nucleolus, where they are required for pre-rRNA processing [50,51]. Given this close physical and functional interaction between the nucleolus and CBs, we wondered if the NS induced in SNI-injured DRG neurons impacts the behaviour of the CB and also if treatment with miR-30c-5p inhibitor influences the organisation of this nuclear compartment.

To determine the organisation and number of CBs under our experimental conditions, we performed immunostaining for coilin—an essential scaffold protein of CBs [45,46]—in dissociated DRG neurons counterstained with PI. CBs appeared as free round coilin-positive nuclear bodies in the nucleoplasm or attached to the nucleolus (Figure 5B,C). The quantitative analysis of CBs in the whole neuronal nucleus showed that the majority of neurons had only one CB, frequently associated with the nucleolus, in all experimental groups (Figure 5B,E,F). Interestingly, a significant increase in the proportion of neurons lacking CBs was detected in SNI-injured neurons at both 5 and 10 days after nerve transection relative to Sham rats (Figure 5A,E,F). Moreover, the treatment of SNI rats with miR-30c-5p mimic further reduced the proportions of neurons lacking CBs (*p* < 0.01), whereas treatment with miR-30c-5p inhibitor prevented it when compared to SNI vehicle rats (Figure 5E,F). Indeed, a significant increase (*p* < 0.001) in the proportion of neurons with more than two CBs (approximately 30%) was estimated in SNI rats treated with miR-30c-5p inhibitor in comparison with untreated SNI animals (Figure 5C–F).

The reorganisation of CBs following the treatment with miR-30c-5p inhibitor was validated by electron microscopy analysis that revealed the presence of some large-size CBs as well as neurons containing more than two CB in a single section of the nucleus (Figure 5G). Given the positive correlation of the number and size of CBs with neuronal activity, previously reported in DRG neurons [35,36], our results suggest a neuroprotective effect of the treatment with the miR-30c-5p inhibitor on the biogenesis of CBs, potentially resulting in enhanced activity of pre-mRNA splicing and mRNA translation.

## 3. Discussion

In this study, we demonstrated the existence of cross-talk between miR-30c-5p function and DRG stress responses after nerve injury, which may contribute to the persistent aberrant signalling in nociception-related pathways underlying neuropathic pain. In particular, we proposed that three essential interconnected cellular responses, the NS (impairment of ribosome biogenesis), CB depletion (dysfunction of pre-mRNA and pre-rRNA processing), and chromatolysis (severe disturbance of mRNA translation), are the pathological mechanisms in stressed DRG neurons underlying neuropathic pain.

Our results revealed that around 25% of the DRG sensory neurons analysed from SNI rats, at days 5 and 10 after SNI, exhibited prominent hallmarks of central chromatolysis. They included (i) NB loss and dissolution in the centre of the cell body, (ii) peripheral displacement of the cell nucleus, (iii) increased number of neurofilaments, and (iv) abundance of small-sized (<0.5 µm) or fragmented mitochondria. Interestingly, in the group of SNI rats treated with miR-30c-5p mimic, the severity of chromatolysis increased significantly (approximately 42% of neurons) in parallel with the intensification of the allodynia developed five days after SNI [40]. This hyperactivation of chromatolysis suggests that a miR-30c-5p-related mechanism is involved in the pathophysiology of neuropathic pain development. Our results are in agreement with previous studies that reported chromatolysis after neural trauma, ischemia, toxicity, and cellular stress [11,13,14,15,16,39,52,53]. It is important to note that the structural and functional changes after neural damage are not identical in all injury types. In fact, the chromatolysis response is dependent on the severity of the damage; it is clearly visible in DRG neurons after the transection of the sciatic nerve but not after crush or chronic constriction injuries [39]. Furthermore, the alterations of DRG neurons reported here are consistent with the hypothesis that chromatolytic disassembly of polyribosomes is induced by endoplasmic reticulum (ER) stress. In this vein, recent studies have demonstrated the activation of ER stress pathways in various neuropathic pain models [53,54].

Ribosomal RNAs are generally stable under physiological conditions, and efficient quality control mechanisms target immature or defective ribosomal particles for endoribonucleolytic cleavage [55]. However, in chromatolytic DRG neurons, the severe loss of ribosomes seems to indicate an NS-dependent deficiency in ribosome biogenesis accompanied by a parallel activation of the endoribonucleolytic degradation of ribosomal subunits by ribonucleases [13]. Curiously, translation silencing in chromatolityc DRG neurons does not induce the formation of cytoplasmic stress granules—which sequester translation initiation factors and some mRNAs released from polyribosome disassembly [56]—as occurs in certain neurodegenerative disorders [28,31]. Moreover, we did not find any ultrastructural features of ribophagy, the specific degradation of ribosomes by autophagy, as appears in several neurodegenerative diseases with chromatolytic neurons [57]. It is noteworthy, however, that the abundance of mitochondria, many of them mini mitochondria (<0.5 µm), in chromatolytic areas of DRG neurons suggests a dysfunction of mitochondrial fusion–fission dynamics. Consistent with this hypothesis, recent studies in neuronal models of chronic stress have reported upregulation of mitochondrial fission via activation of Drp1 (dynamin-related protein), resulting in increased formation of dysfunctional mini mitochondria and impairment in energy metabolism [58].

Interestingly, treatment with miR-30c-5p inhibitor largely prevented the chromatolytic reaction of the injured DRG neurons and, in parallel, protected rats against allodynia development 10 days after SNI [40,41]. Thus, in SNI rats treated with miR-30c-5p inhibitor, most neurons exhibited a normal organisation of the nucleolus and NBs, similar to that of control sham-operated rats.

In fact, the presence of chromatolytic neurons was significantly reduced (approximately 10%). In the event of severe injury, neurons may remain chromatolytic and fail to restore normal protein synthesis, leading to atrophy or death [11,13]. It is, however, remarkable that, under conditions of severe chromatolysis from a variety of stressors, healthy nucleoli preserve their normal configuration and activate their transcriptional activity as a compensatory response to promote functional recovery [12,34,39]. In this context, it seems reasonable to suggest that the targeting of miR-30c-5p may result in an improvement in the survival rate of neurons following axotomy. Nevertheless, further analysis is required to confirm this hypothesis.

Potential mechanisms by which miR-30c-5p gain of function may contribute to the exacerbation of chromatolysis include the activation of catabolic pathways mediated by ribonucleases, as commented above [13] and the NS-dependent impairment of ribosome biogenesis [19,22,23,27,31]. The nucleolus plays a crucial role in maintaining the structural and functional integrity of the protein synthesis machinery by fine-tuning ribosome biogenesis to meet the demands of cellular translation [22,35]. The morphology and size of nucleoli are linked to nucleolar activity, which is inevitably altered under stress conditions, showing a variety of reorganisation features [23,25,26,27,28,30,31,57]. Our results support the induction of an NS response in DRG neurons from SNI rats, which is aggravated by miR-30c-5p mimic treatment. It is well known that FCs, with their associated DFC shell, constitute nucleolar transcription factories for rRNA synthesis [18,20,21]. Accordingly, the number of FCs positively correlates with cell body size and transcriptional activity in DRG neurons [35]. By using immunolabelling for UBF, a marker of FCs, we demonstrated a drastic reorganisation of the numerous small-sized FCs of control neurons (sham-operated rats) into a giant segregated FC. This finding indicates a severe reduction in the number of nucleolar transcription factories, supporting an inhibition of ribosome biogenesis. Indeed, the presence of only one large-sized FC occurs in cells with a very low rate of ribosome biogenesis, such as resting lymphocytes and cerebellar granule cells [19,44]. Another important component of the NS response in chromatolytic DRG neurons is the peripheral segregation of large masses of GC and DFC, the latter forming typical “nucleolar caps”. Nucleolar segregation has been reported during transcriptional inhibition under different cellular stresses, such as treatment with genotoxic agents or UV irradiation [22,23], as well as in experimental models of neurodegeneration [27,30,31,48]. Finally, a recent study reported that the nucleolar accumulation of certain ribosome-free ribosomal proteins induced NS with a reduced rate of ribosome biogenesis [24]. This raises the possibility that proteins released from ribosome disassembly in chromatolytic neurons may contribute to NS induction NS in DRG neurons. In this context, the NS response is emerging as an important sensor of neuronal dysfunction in several neurodegenerative and neurodevelopmental disorders [25,26,27,28,29,30,31,32,33].

Previous investigations reported the involvement of members of the miR-30 family in RER and NS responses, as well as subsequent apoptosis following cell damage [59,60]. It is also noteworthy that members of the miR-30 family can interact with nucleophosmin, a nucleolar chaperone involved in RER stress and nuclear disintegration [48]. Moreover, ribosome biogenesis is a complex process regulated at multiple steps by miRNAs [61]. It is important to note that, in addition to their post-transcriptional regulatory effects, miRNAs also have specific nuclear functions [62]; several miRNAs are significantly enriched in the nucleolus [63,64,65]. These nucleolar miRNAs may have non-canonical biological roles, including the regulation of rRNA synthesis and ribosomal subunit assembly [63,66] and the protected formation of pre-silenced miRNA, i.e., mRNA pairs before they are exported to the cytoplasm [67]. Alternatively, nucleoli may serve as the storage site for miRNAs that undergo dynamic changes in response to genotoxic stress as a defensive response. Subsequently, they are redistributed to the nucleoplasm and/or cytoplasm to maintain genome integrity [64]. Furthermore, computational analysis has identified microRNA-mediated control of ribosomal proteins as a crucial potentiator of ribosome biogenesis activity and disease progression [68]. A very recent study reported that several miRNAs play pivotal roles as negative regulators of ribosome biogenesis. The overexpression of these miRNAs has been observed to induce a robust repression of nucleolar rRNA biogenesis in human cell lines [69]. Given the evidence that miR-30c is significantly concentrated in the nucleolus [65], it is plausible that this miRNA plays a nucleolar role in ribosome biogenesis after nerve injury. Collectively, these results provide compelling evidence that miR-30c-5p gain of function enhances the NS response after axotomy, rendering the cell unable to cope with axotomy-induced RER disruption and chromatolysis.

The CB is a multifunctional nuclear organelle mainly involved in the biogenesis of snRNPs and snoRNPs required for pre-mRNA and pre-rRNA processing, respectively [45,46,50]. CBs are physically and functionally linked to the nucleolus. A paradigmatic example of CB and nucleolus cooperation is the assembly in the CB of the snoRNPs required for rRNA processing within the DFC of the nucleolus [45,49,50,51]. Therefore, we investigated the potential impact of miR-30c-5p modulation on CBs in DRG neurons from SNI rats.

Our results showed an increase in the proportion of DRG neurons lacking CBs in both SNI and SNI treated with miR-30c-5p mimic rats. In contrast, there was a significant increase in the percentage of neurons with more than two CBs in the DRG from SNI rats treated with miR-30c-5p inhibitor. Moreover, electron microscopy analysis revealed the presence of large-sized CBs. In different neuronal types, it is well established that CB size and number are positively correlated with global transcriptional activity and cellular demand of spliceosomal snRNPs and nucleolar snoRNPs for pre-mRNA and pre-rRNA processing, respectively [22,45,46]. In this context, we proposed that CBs were not disrupted by SNI-related stress and that changes in CB number might be to adapt CB biogenesis to the different nuclear (pre-mRNA) and nucleolar (pre-rRNA) transcriptional activities of DRG neurons under each experimental condition. Therefore, the reduced rate of ribosome biogenesis and translational inhibition in chromatolytic neurons of miR-30c-5p-treated SNI rats was associated with decreased CB numbers. Conversely, treatment with miR-30c-5p inhibitor would preserve CB biogenesis following SNI.

## 4. Materials and Methods

### 4.1. Animals

The experiments were performed in eight-week-old (250 to 300 g) male Sprague–Dawley rats. All animals were kept under controlled conditions of humidity and temperature (22 ± 1 °C, 60–70% relative humidity) on a 12 h light/dark cycle (lights on at 8:00 a.m.). Food and water were supplied ad libitum. Under anaesthesia with isoflurane, animals were sacrificed by decapitation or perfused with a freshly prepared solution containing 3.7% paraformaldehyde (Merk, Madrid, Spain) in HEPEM buffer (2xHPEM: HEPES 60 mM, Pipes 130 mM, EGTA 20 mM, and MgCl2·6H_2_O 4 mM, pH 6.9) containing 0.5% Triton X-100 through the left ventricle using a 30G needle, according to the experimental procedure. The study was approved by the University of Cantabria Institutional Laboratory Animal Care and Use Committee (reference IP1020) and conducted following the guidelines of directive 2010/63/EU of the European Parliament and the International Association for the Study of Pain.

#### 4.1.1. Neuropathic Pain Model

Chronic neuropathic pain was induced using the spared nerve injury (SNI) model [70]. Briefly, rats were anaesthetised with isoflurane (Forane^®^, 1.5–2.5%, 30% N20, and 70% O_2_), and the left common sciatic nerve was exposed at the level of its trifurcation. The tibial and common peroneal branches of the nerve were sectioned; the sural nerve was left untouched. Sham-operated rats underwent the same procedure, but all sciatic branches were left intact.

#### 4.1.2. Treatment and Experimental Design

The treatments were administered in the cisterna magna dissolved in a mixture of lipofectamine (Life Technologies, Invitrogen, Waltham, MA, USA) and artificial CSF. Rats were anaesthetised with isoflurane and placed on a stereotaxic frame for intracisternal administration of miR-30c-5p mimic (mirVana MC11060, Thermo Fisher Science, Waltham, MA, USA), miR-30c-5p inhibitor (mirVana MH11060, Thermo Fisher Science, Waltham, MA, USA), or vehicle. The following experimental designs were used based on our previous results. The administration of a miR-30c-5p mimic accelerates the development of allodynia. However, the administration of a miR-30c-5p inhibitor delayed neuropathic pain development [40,41].

Experimental design 1. Rats received a cycle of three intracisternal injections of miR-30c-5p mimic (100 ng/10 µL) at the time of the SNI or sham intervention and on days 2 and 4 after surgery.

Experimental design 2. Rats received a cycle of three intracisternal injections of miR-30c-5p inhibitor (100 ng/10 µL) at the time of the SNI or sham intervention and on days 4 and 7 after surgery.

Sham animals received intracisternal injections composed of artificial CSF and lipofectamine following identical protocols.

Rats were sacrificed on day 5 or 10 after nerve injury or sham intervention, and DRGs (L3, L4, and L5) were dissected and stored until further use.

### 4.2. Microscopy Techniques

#### 4.2.1. Immunofluorescence and Confocal Microscopy

Rats were anaesthetised and perfused with paraformaldehyde 3.7% in phosphate-buffered saline (PBS), and lumbar DRGs (L3, L4, and L5) were collected and washed three times in PBS. DRG neuron dissociates were obtained in squash preparations as previously described [39].

For the immunofluorescence studies, squash preparations were sequentially treated with 0.1 glycine in PBS for 20 min, 0.5% Triton X-100 in PBS for 45 min, and 0.05% Tween20 in PBS for 5 min. Samples were then incubated overnight at 4 °C with the primary antibody, washed with 0.05% Tween20 in PBS, and incubated for 45 min in the specific secondary antibody conjugated with FITC or TexasRed (Jackson, Laboratories, USA Jackson, West Grove, PA, USA). To analyse the distribution and integrity of NBs, some samples were stained with PI for 15 min at room temperature. Finally, the preparations were mounted with Vectashield with DAPI^®^ (Vector Laboratories, Inc., Burlingame, CA, USA), and confocal images were obtained with an LSM510 laser confocal microscope (Zeiss, Oberkochen, Germany) using a 63× oil (1.4 NA) objective. To avoid overlapping signals, images were obtained by sequential excitation at 488 nm and 543 nm to detect FITC or TexasRed, respectively.

The following primary antibodies were used: monoclonal mouse anti-UBF (1:100; sc-13125, Santa Cruz Biotechnology, Santa Cruz, CA, USA), polyclonal rabbit anti-Lamb1 (1:100; ab108536, Abcam), and polyclonal rabbit anti-coilin 204/10 (1:300; 204.3 serum).

The quantitative analysis was performed using ImageJ software 1.x (NIH, Bethesda, MD, USA; https://imagej.net/, accessed on 15 March 2020). At least 1000 neurons per animal were examined (n = 3 animals per experimental condition). For the quantitative analysis of the number of CBs and nucleoli, coilin and UBF-positive spots were counted, respectively, on serial confocal sections of the whole nucleus. Images were processed using Adobe Photoshop CS4 software (Adobe Systems Inc., v 20.0.0, San Jose, CA, USA).

#### 4.2.2. Electron Microscopy

For conventional ultrastructural examination of DRG neurons, rats were perfused under deep anaesthesia with 3% glutaraldehyde in 0.1 M phosphate buffer, pH 7.4. DRGs were removed and post-fixed for approximately 1 h with the perfusion buffer at room temperature. After several 15 min washes in 0.12 M phosphate buffer, the microdissected DRG were post-fixed in 2% osmium tetroxide, dehydrated in increasing concentrations of acetone, and then embedded in Araldite (Durcupan, Fluka, Switzerland). Ultrathin sections mounted on copper grids were stained with lead citrate and uranyl acetate and examined using an electron microscope (Philips EM 208, Amsterdam, The Netherlands). Images were taken with an optical microscope (AxiosKop2 plus, Jena, Germany) coupled to a video camera (AxioCam HRC Zeiss, Jena, Germany).

### 4.3. Statistical Analysis

GraphPad Prism 5.01, Predictive Analytics SoftWare (PASW) 22 [Statistical Package for the Social Sciences (SPSS) Inc., IBM, New York, NY, USA], and Stata 14/SE (StataCorp, College Station, TX, USA) packages were used. Rat data were expressed as means ± standard error of the mean (SEM). Differences between multiple groups were analysed by one-way or Two-way ANOVA. The statistical values and variables analysed for each result are presented in a table in the Appendix A.

## Figures and Tables

**Figure 1 ijms-25-11427-f001:**
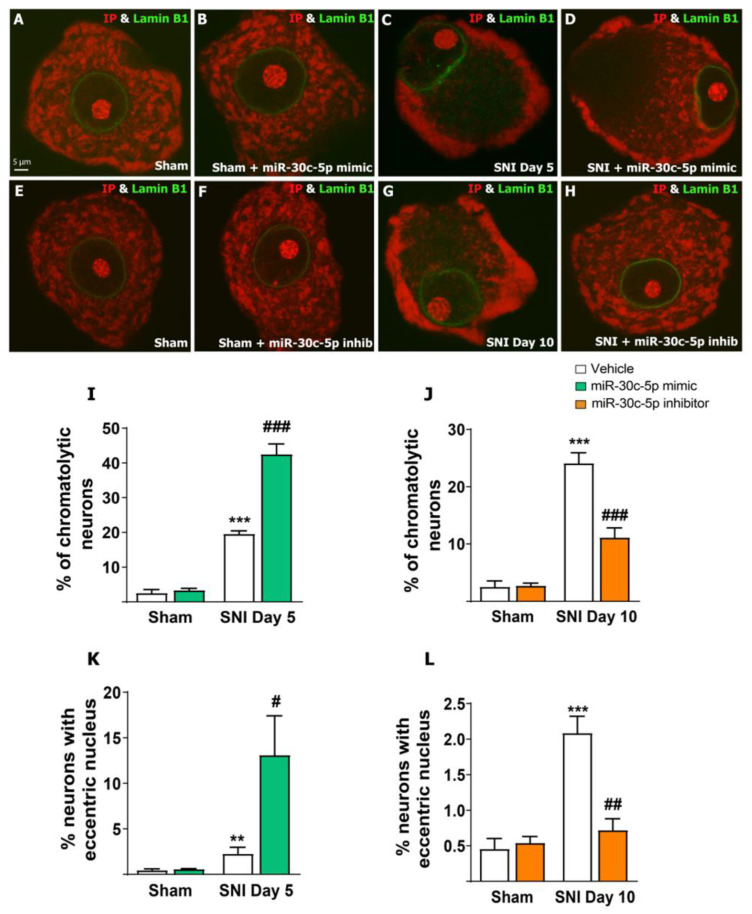
miR-30c-5p modulation effects on the chromatolysis developed by dorsal root ganglia neurons after spared nerve injury. (**A**–**H**) Dissociated dorsal root ganglia (DRG) neurons double stained with propidium iodide (PI, red) and Lamin B1 (green). Note the prominent NBs and round nuclei in sham rats treated with vehicle (**A**,**E**), miR-30c-5p mimic (**B**), or miR-30c-5p inhibitor (**F**), reflecting a normal distribution of the protein synthesis machinery and nuclear location. DRG neurons from day-5 (**C**) or day-10 SNI rats (**G**) exhibited central chromatolysis with dispersion and severe loss of NBs in the centre of the neuronal body, accumulations of Nissl substance at the marginal cytoplasm, and peripheral displacement of the nucleus, which were aggravated by treatment with miR-30c-5p mimic (**D**). Administration of miR-30c-5p inhibitor reduced the chromatolytic response observed after SNI (**H**). (**I**,**J**) Percentage of neurons showing chromatolysis. (**K**,**L**) Percentage of neurons showing eccentricity of the nucleus. The percentage of damaged neurons and eccentric nuclei was determined in 1000 neurons per rat (n = 3 rats per group). ** *p* < 0.01, *** *p* < 0.001 vs. Sham; # *p* < 0.05, ## *p* < 0.01, ### *p* < 0.001 vs. SNI (Two-way ANOVA followed by the Bonferroni post hoc test). Scale bar: 5 µm.

**Figure 2 ijms-25-11427-f002:**
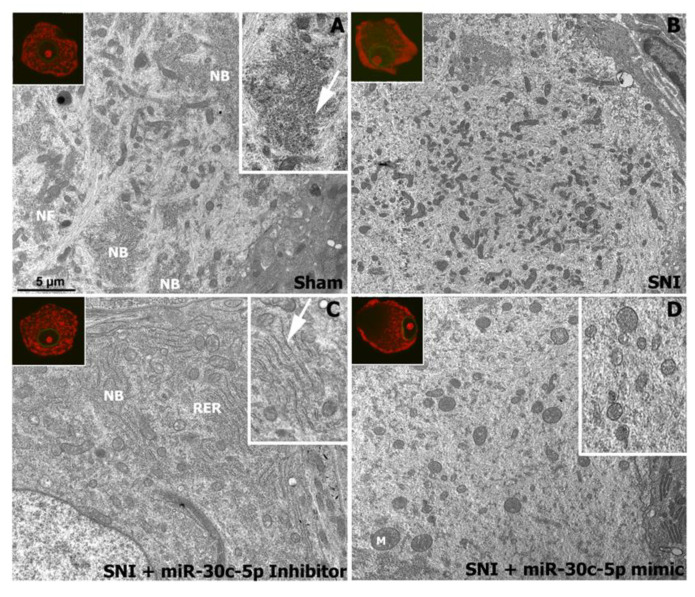
Electron micrographs illustrating the ultrastructural characteristics of dorsal root ganglia neurons after administration of miR-30c-5p mimic or inhibitor to SNI rats. In dorsal root ganglia (DRG) neurons from sham (**A**) and SNI rats treated with miR-30c-5p inhibitor (**C**), the most prominent organelles are the NBs, composed of RER cisterns (**C**, arrow) and rosettes of free polyribosomes (**A**, arrow). Bundles of neurofilaments (NF) interspersed between NBs, profiles of Golgi complexes, and mitochondria are also apparent. In DRGs from SNI rats treated with vehicle (**B**) or miR-30c-5p mimic (**D**), the NBs disaggregated, leaving an extensive cleared chromatolytic area in the centre of the cell body, free of NBs. The increased number of NFs and the abundance of mitochondria (M)—some of which are very small (<0.5 µm)—in chromatolytic areas are also noteworthy. Scale bar: 5 µm.

**Figure 3 ijms-25-11427-f003:**
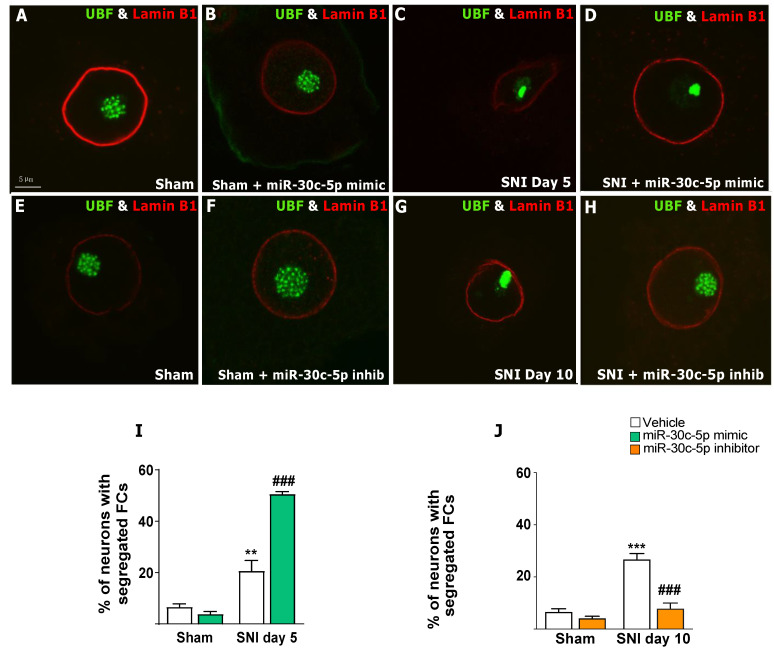
miR-30c-5p modulation effects on the nucleolar organisation of dorsal root ganglion neurons after spared nerve injury. (**A**–**H**) Dissociated dorsal root ganglia (DRG) neurons double immunostained for upstream binding factor (UBF, green) and Lamin B1 (red). DRG neurons from sham rats treated with vehicle (**A**,**E**), miR-30c-5p mimic (**B**), or miR-30c-5p inhibitor (**F**), and day-10 SNI rats treated with miR-30c-5p inhibitor (**H**) presented a normal UBF distribution as small dots corresponding to FCs. In contrast, DRG neurons from day-5 SNI rats treated with vehicle (**C**) or miR-30c-5p mimic (**D**) and day-10 SNI rats treated with vehicle (**G**) showed segregation of UBF nucleolar staining into one or a few giant FCs. (**I**,**J**) The percentage of neurons showing UBF-positive giant FCs was determined in 1000 neurons per rat (n = 3 rats per group); ** *p* < 0.01, *** *p* < 0.001 vs. Sham; ### *p* < 0.001 vs. SNI (Two-way ANOVA followed by the Bonferroni post hoc test). Scale bar: 5 µm.

**Figure 4 ijms-25-11427-f004:**
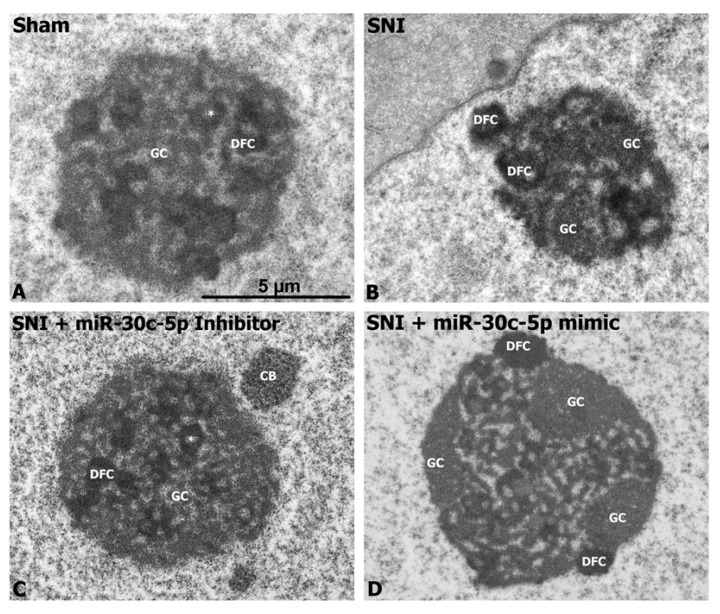
Representative electron micrographs illustrating the ultrastructural nucleolar characteristics of dorsal root ganglia neurons after administration of miR-30c-5p mimic or inhibitor to SNI rats. Sham (**A**) and SNI rats treated with miR-30c-5p inhibitor (**C**) exhibit the typical nucleolar organisation of DRG neurons, characterised by the presence of numerous small-sized fibrillar centres (*, FCs), surrounded by a ring of dense fibrillar component (DFC), and areas of granular component (GC), preferentially at the nucleolar periphery. SNI rats treated with vehicle (**B**) or with miR-30c-5p mimic (**D**) present severe nucleolar alterations, including the formation of enlarged FCs and segregation of large masses of GC and DFC at the nucleolar periphery. Scale bar: 2 µm.

**Figure 5 ijms-25-11427-f005:**
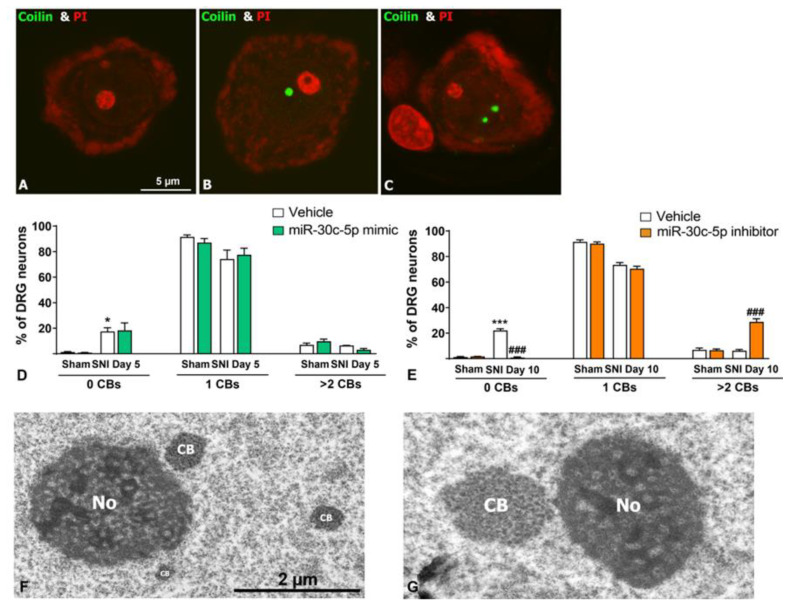
miR-30c-5p modulation effects on the number of Cajal bodies in dorsal root ganglion neurons after spared nerve injury. Representative images of dissociated DRG neurons immunolabeled for coilin (green) and counterstained with propidium iodide ((PI), red). Example of neurons showing 0 (**A**), 1 (**B**), and 2 (**C**) CBs. (**D**,**E**) Quantitative analysis of the percentage of neurons carrying 0, 1, or more than 2 CBs in each of our experimental groups. The number of CBs per neuron was determined in 1000 neurons per rat, in 3 rats of each group (sham; SNI + vehicle; SNI + miR-30c-5p inhibitor; SNI + miR-30c-5p mimic). The quantification analysis indicates that, regardless of the experimental condition, most neurons present 1 CB. There is a significant increase in the percentage of neurons showing more than 2 CBs in SNI rats treated with miR-30c-5p inhibitor. The proportion of neurons without CBs is significantly increased in SNI rats treated with vehicle or miR-30c-5p mimic. (**F**,**G**) Electron microscopy of CBs in DRG neurons from SNI rats treated with miR-30c-5p inhibitor showing 3 CBs (**F**) and a hypertrophic CB physically close to the nucleolus (**G**). * *p* < 0.05, *** *p* < 0.01 vs. Sham; ### *p* < 0.001 vs. SNI). (Two-way ANOVA followed by the Bonferroni post hoc test). Scale bar: 5 µm. Scale bar: 2 µm.

## Data Availability

The data presented in this study are available in the article.

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
