# Peer review of "miR-30c-5p Gain and Loss of Function Modulate Sciatic Nerve Injury-Induced Nucleolar Stress Response in Dorsal Root Ganglia Neurons"

_ijms, 2024, doi:10.3390/ijms252111427_

Round 1
Reviewer 1 Report
Comments and Suggestions for Authors
The authors of this article provide new data about the implications of miR-30c-5p gain and loss of function modulated in the SNI model of neuropathic pain. The article is well structured and fairly well written. However, some aspects that could improve the quality have been detected as follows starting by major considerations:
1. Why do you use two experimental designs? (510-513)
“Experimental design 1. Rats received a cycle of three intracisternal injections of miR- 509 30c-5p mimic (100 ng/10 µL) at the time of the SNI or sham intervention and on days 2 and 4 after surgery.
Experimental design 2. Rats received a cycle of three intracisternal injections of miR- 30c-5p inhibitor (100 ng/10 µL) at the time of the SNI or sham intervention and on days 4 and 7 after surgery.”
And then you sacrificed the rats on day 5 or 10 after nerve injury or sham intervention?
2. In the fig. 1 and 3, you mention n=3 rats per group and in the Statistical analysis you mention that “Differences between multiple groups were analyzed by one-way or two-way ANOVA”. Why do you use this small sample size (n=3)?
ANOVA has specific assumptions, including normality of the data and homogeneity of variances. With such a small sample size, it can be challenging to meet these assumptions, and the power of the test may be limited.
To support the results by this statistical test, it is necessary to supplementing with descriptive statistics and effect sizes.
From the two aspects mentioned above, if the number of animals in the group is increased to n= 5 and if you use only a single experimental design with administration at 2, 4 and 7 days after surgery, the results would have been firmly supported by the ANOVA test.
Author Response
Response to Reviewer 1 Comments
The authors of this article provide new data about the implications of miR-30c-5p gain and loss of function modulated in the SNI model of neuropathic pain. The article is well structured and fairly well written. However, some aspects that could improve the quality have been detected as follows starting by major considerations:
Many thanks for your valuable comments and help us improve our manuscript.
Comments 1: Why do you use two experimental designs? (510-513)
Experimental design 1. Rats received a cycle of three intracisternal injections of miR-30c-5p mimic (100 ng/10 µL) at the time of the SNI or sham intervention and on days 2 and 4 after surgery.
Experimental design 2. Rats received a cycle of three intracisternal injections of miR- 30c-5p inhibitor (100 ng/10 µL) at the time of the SNI or sham intervention and on days 4 and 7 after surgery.
And then you sacrificed the rats on day 5 or 10 after nerve injury or sham intervention?
Response 1: Following the reviewer’s query, a more detailed description of the protocols used in the study has been added in the revised version of the manuscript (page 13, lines 491 to 493).
The rationale behind the choice of these two experimental designs is based on the findings of previous studies conducted by our research group. We previously demonstrated a relation between miR-30c-5p and neuropathic pain in rats subjected to spared nerve injury (SNI) (Tramullas et al., 2018). The results of this study demonstrate that miR-30c-5p is up-regulated in the spinal dorsal horn, dorsal root ganglia, cerebrospinal fluid and plasma in rats exposed to SNI, and the expression of miR-30c-5p positively correlates with the severity of the allodynia developed after SNI. Furthermore, the overexpression and silencing of miR-30c-5p exhibited opposing effects on the development of allodynia following nerve injury. The administration of a cycle of three intracisternal injections of an miR-30c-5p mimic into the cisterna magna at the time of the sciatic nerve injury or sham intervention and on days 2 and 4 after surgery accelerates the development of allodynia. However, the administration of a cycle of three intracisternal injections of miR-30c-5p inhibitor at the time of the sciatic nerve injury or sham intervention and on days 4 and 7 after surgery delayed neuropathic pain development.
According to these two protocols, rats were sacrificed on day 5 or 10 after nerve injury or sham intervention, when the greatest differences were observed in the degree of mechanical allodynia developed between groups.
Comments 2: In the fig. 1 and 3, you mention n=3 rats per group and in the Statistical analysis you mention that “Differences between multiple groups were analysed by one-way or two-way ANOVA”. Why do you use this small sample size (n=3)?
ANOVA has specific assumptions, including normality of the data and homogeneity of variances. With such a small sample size, it can be challenging to meet these assumptions, and the power of the test may be limited.
To support the results by this statistical test, it is necessary to supplementing with descriptive statistics and effect sizes.
Response 2: We agree with this comment and following the reviewer’s recommendation and to support our results despite the small sample (n=3), we have now added in the supplementary section a more detailed statistical analysis with the following:
- Descriptive statistics: We have included means and standard deviations for each group to provide a clearer understanding of the data distribution.
- Effect sizes: We have recalculated the data to include effect sizes (eta-squared) to provide additional insight into the magnitude of the differences between groups through the quantification of the proportion of total variance explained by one or more independent variables.
It is important to note that the eta-squared values in our study range from 0.43 to 0.97, indicating that the independent variable has a considerable impact on the dependent variable.
Comments 3: From the two aspects mentioned above, if the number of animals in the group is increased to n= 5 and if you use only a single experimental design with administration at 2, 4 and 7 days after surgery, the results would have been firmly supported by the ANOVA test.
Response 3: Thank you for your comment. However, as we explained in response to comments 1 and 2, our study design is based on previous results and to determine the effect of miR-30c-5p modulation in vivo on chromatolysis and nucleolar stress, the most appropriate experimental designs are as we described above. Furthermore, although the sample size is small, the one- or two-way ANOVA analysis is appropriate and supports our results given the eta-squared values.

Reviewer 2 Report
Comments and Suggestions for Authors
In this paper, the authors aim to elucidate the role of miR-30c-5p in the development of neuropathic pain, focusing on the ultrastructural changes of DRG neurons following a SNI. The study is continuation of a previous study in which they found that administration of a miR-30c mimic accelerated the development of neuropathic pain. In this sense, its novelty is limited. However, the quality of the microscopic methods used is excellent, and the new results add valuable knowledge to the topic.
Comments:
- In the SNI the DRG neurons projecting axons along the sural nerve remain intact; therefore, though small, a proportion of the studied neurons are not affected. It would have been better to use the complete transection of the sciatic nerve model for this type of study. Please comment on the limitation that the SNI model imposes.
- The plots of % neurons with chromatolysis and eccentric nucleus shown in Fig 1 I-L are confusing, and appear incomplete. You should make the same plot for the six conditions (Sham or SNI treated with vehicle, miR-30c-5p mimic or inhibitor) foe either 5 or 10 days after operation together and make the appropriate statistical analysis.
- The reason why samples were taken 5 days after surgery for the miR-30c-5p mimic and at 10 days for the inhibitor should be explained. In addition, evidence that SNI causes similar quantitative changes in DRG ultrastructural characteristics at 5 and 10 days could be provided.
- For multiple comparisons which post-hoc test was used after ANOVA?
- The Discussion may be shortened. The first two paragraphs are repeated from the Introduction, and can be eliminated.
Author Response
Response to Reviewer 2 Comments
In this paper, the authors aim to elucidate the role of miR-30c-5p in the development of neuropathic pain, focusing on the ultrastructural changes of DRG neurons following a SNI. The study is continuation of a previous study in which they found that administration of a miR-30c mimic accelerated the development of neuropathic pain. In this sense, its novelty is limited. However, the quality of the microscopic methods used is excellent, and the new results add valuable knowledge to the topic.
Many thanks for your valuable comments and help us improve our manuscript.
Comments 1: In the SNI the DRG neurons projecting axons along the sural nerve remain intact; therefore, though small, a proportion of the studied neurons are not affected. It would have been better to use the complete transection of the sciatic nerve model for this type of study. Please comment on the limitation that the SNI model imposes.
Response 1: Following the reviewer’s query, we have now added a more detailed description of the limitation that the SNI model presented in the introduction (page 3, lines 99 to 104) and discussion (page 11, lines 351 to 355) section of the revised version of the manuscript.
It is important to note that the structural and functional changes after neural damage are not identical in all injury types. Indeed, the chromatolysis response is dependent on the severity of the damage; it is clearly visible in DRG neurons after the transection of the sciatic nerve but not after crush or chronic constriction injuries (Delibas et al., 2024). In our study, we have used the spared nerve injury, in which, the tibial and common peroneal branches of the nerve are sectioned, leaving the sural branch intact. Therefore, the anticipated degree of chromatolysis may be less pronounced in this model than in a complete transection of the sciatic nerve. Notably, approximately 25% of the DRG sensory neurons analysed from SNI rats at days 5 and 10 after SNI exhibited prominent hallmarks of central chromatolysis, indicating that not all neurons were affected.
Furthermore, the election of this model was accurate because in our study we aimed to address the relationship between structural changes of DRG neurons following nerve injury and the subsequent development of neuropathic pain. By using the transaction model, the evaluation of pain would have been not possible.
Comments 2: The plots of % neurons with chromatolysis and eccentric nucleus shown in Fig 1 I-L are confusing, and appear incomplete. You should make the same plot for the six conditions (Sham or SNI treated with vehicle, miR-30c-5p mimic or inhibitor) foe either 5 or 10 days after operation together and make the appropriate statistical analysis.
Response 2: In response to the reviewer's comments, we have modified Figure 1 (I-L) to provide a clearer representation of the results. In each graph, the four experimental conditions are now plotted according to each experimental design, following the reviewer's request. Graphs I and K illustrate the data for the sham, sham + mir 30c-5p mimic, SNI and SNI + mir 30c-5p mimic groups, while graphs J and L depict the data for the sham, sham + mir 30c-5p inhibitor, SNI and SNI + mir 30c-5p inhibitor groups. However, the authors consider it inappropriate to perform a statistical analysis combining all groups due to the aforementioned reasons related to the experimental design. A comprehensive statistical analysis has been included in the supplementary materials.
Comments 3: The reason why samples were taken 5 days after surgery for the miR-30c-5p mimic and at 10 days for the inhibitor should be explained. In addition, evidence that SNI causes similar quantitative changes in DRG ultrastructural characteristics at 5 and 10 days could be provided.
Response 3: Following the reviewer’s query, a more detailed description of the protocols used in the study has been added in the revised version of the manuscript (page 13, lines 491 to 493).
The rationale behind the choice of these two experimental designs is based on the findings of previous studies conducted by our research group. We previously demonstrated a relation between miR-30c-5p and neuropathic pain in rats subjected to spared nerve injury (SNI) (Tramullas et al., 2018). The results of this study demonstrate that miR-30c-5p is up-regulated in the spinal dorsal horn, dorsal root ganglia, cerebrospinal fluid and plasma in rats exposed to SNI, and the expression of miR-30c-5p positively correlates with the severity of the allodynia developed after SNI. Furthermore, the overexpression and silencing of miR-30c-5p exhibited opposing effects on the development of allodynia following nerve injury. The administration of a cycle of three intracisternal injections of an miR-30c-5p mimic into the cisterna magna at the time of the sciatic nerve injury or sham intervention and on days 2 and 4 after surgery accelerates the development of allodynia. However, the administration of a cycle of three intracisternal injections of miR-30c-5p inhibitor at the time of the sciatic nerve injury or sham intervention and on days 4 and 7 after surgery delayed neuropathic pain development. In accordance with these two protocols, rats were sacrificed on day 5 or 10 after nerve injury or sham intervention, when the greatest differences were observed in the degree of mechanical allodynia developed between groups.
In agreement with the reviewer’s comment, we have now added a statistical analysis in the results section to demonstrate that there were no significant differences when comparing day-5 SNI rats with day-10 SNI rats in the ultrastructural changes in DRG neurons (page 3, lines 143 to 145 and page 7, lines 227 to 228). Furthermore, a detailed statistical analysis has also been added to the Supplementary section.
Comments 4: For multiple comparisons which post-hoc test was used after ANOVA?
Response 4: We have used the Bonferroni post-hoc test.
Comments 5: The Discussion may be shortened. The first two paragraphs are repeated from the Introduction, and can be eliminated.
Response 5: Following the reviewer's suggestion, we have removed the first two paragraphs of the Discussion.

Round 2
Reviewer 1 Report
Comments and Suggestions for Authors
The comments and changes made in the article answered the questions formulated previously. The justifications are reasoned. The article can be published.